# ACTIVE CONTRASTIVE LEARNING OF AUDIO-VISUAL VIDEO REPRESENTATIONS

**Shuang Ma**[*]
Microsoft
Redmond, WA, USA

**Zhaoyang Zeng**[*]
Sun Yat-sen University
Guangzhou, China

**Daniel McDuff**
Microsoft Research
Redmond, WA, USA

**Yale Song**
Microsoft Research
Redmond, WA, USA

## ABSTRACT

Contrastive learning has been shown to produce generalizable representations of audio and visual data by maximizing the lower bound on the mutual information (MI) between different views of an instance. However, obtaining a tight lower bound requires a sample size exponential in MI and thus a large set of negative samples. We can incorporate more samples by building a large queue-based dictionary, but there are theoretical limits to performance improvements even with a large number of negative samples. We hypothesize that *random negative sampling* leads to a highly redundant dictionary that results in suboptimal representations for downstream tasks. In this paper, we propose an active contrastive learning approach that builds an *actively sampled* dictionary with diverse and informative items, which improves the quality of negative samples and improves performances on tasks where there is high mutual information in the data, e.g., video classification. Our model achieves state-of-the-art performance on challenging audio and visual downstream benchmarks including UCF101, HMDB51 and ESC50.[1]

## 1 INTRODUCTION

Contrastive learning of audio and visual representations has delivered impressive results on various downstream scenarios (Oord et al., 2018; Hénaff et al., 2019; Schneider et al., 2019; Chen et al., 2020). This self-supervised training process can be understood as building a dynamic dictionary per mini-batch, where "keys" are typically *randomly* sampled from the data. The encoders are trained to perform dictionary look-up: an encoded "query" should be similar to the value of its matching key and dissimilar to others. This training objective maximizes a lower bound of mutual information (MI) between representations and the data (Hjelm et al., 2018; Arora et al., 2019). However, such lower bounds are tight only for sample sizes exponential in the MI (McAllester & Stratos, 2020), suggesting the importance of building a large and consistent dictionary across mini-batches.

Recently, He et al. (2020) designed Momentum Contrast (MoCo) that builds a queue-based dictionary with momentum updates. It achieves a large and consistent dictionary by decoupling the dictionary size from the GPU/TPU memory capacity. However, Arora et al. (2019) showed that simply increasing the dictionary size beyond a threshold does not improve (and sometimes can even harm) the performance on downstream tasks. Furthermore, we find that MoCo can suffer when there is high redundancy in the data, because only relevant – and thus limited – parts of the dictionary are updated in each iteration, ultimately leading to a dictionary of redundant items (we show this empirically in Fig. 3). We argue that *random negative sampling* is much responsible for this: a randomly constructed dictionary will contain more "biased keys" (similar keys that belong to the same class) and "ineffective keys" (keys that can be easily discriminated by the current model) than a carefully constructed one. Furthermore, this issue can get aggravated when the dictionary size is large.

In this paper, we focus on learning audio-visual representations of video data by leveraging the natural correspondence between the two modalities, which serves as a useful self-supervisory signal (Owens & Efros, 2018; Owens et al., 2016; Alwassel et al., 2019). Our starting point is contrastive learning (Gutmann & Hyvärinen, 2010; Oord et al., 2018) with momentum updates (He et al., 2020).

---

[*]Equal contribution
[1]Code is available at: `https://github.com/yunyikristy/CM-ACC`

However, as we discussed above, there are both practical challenges and theoretical limits to the dictionary size. This issue is common to all natural data but is especially severe in video; successive frames contain highly redundant information, and from the information-theoretic perspective, audio-visual channels of video data contain higher MI than images because the higher dimensionality – i.e., temporal and multimodal – reduces the uncertainty between successive video clips. Therefore, a dictionary of *randomly sampled* video clips would contain highly redundant information, causing the contrastive learning to be ineffective. Therefore, we propose an *actively sampled* dictionary to sample informative and diverse set of negative instances. Our approach is inspired by active learning (Settles, 2009) that aims to identify and label only the maximally informative samples, so that one can train a high-performing classifier with minimal labeling effort. We adapt this idea to construct a non-redundant dictionary with informative negative samples.

Our approach, Cross-Modal Active Contrastive Coding (CM-ACC), learns discriminative audio-visual representations and achieves substantially better results on video data with a high amount of redundancy (and thus high MI). We show that our *actively sampled* dictionary contains negative samples from a wider variety of semantic categories than a randomly sampled dictionary. As a result, our approach can benefit from large dictionaries even when randomly sampled dictionaries of the same size start to have a deleterious effect on model performance. When pretrained on AudioSet (Gemmeke et al., 2017), our approach achieves new state-of-the-art classification performance on UCF101 (Soomro et al., 2012), HMDB51 (Kuehne et al., 2011), and ESC50 (Piczak, 2015b).

## 2 BACKGROUND

**Contrastive learning** optimizes an objective that encourages similar samples to have similar representations than with dissimilar ones (called negative samples) (Oord et al., 2018):

$$\min_{\theta_f, \theta_h} \mathbb{E}_{x \backsim p_\mathcal{X}} \left[ -log \left( \frac{e^{f(x;\theta_f)^\intercal h(x^+;\theta_h)}}{e^{f(x;\theta_f)^\intercal h(x^+;\theta_h)} + e^{f(x;\theta_f)^\intercal h(x^-;\theta_h)}} \right) \right] \tag{1}$$

The samples $x^+$ and $x^-$ are drawn from the same distribution as $x \in \mathcal{X}$, and are assumed to be similar and dissimilar to $x$, respectively. The objective encourages $f(\cdot)$ and $h(\cdot)$ to learn representations of $x$ such that $(x, x^+)$ have a higher similarity than all the other pairs of $(x, x^-)$.

We can interpret it as a dynamic dictionary look-up process: Given a "query" $x$, it finds the correct "key" $x^+$ among the other irrelevant keys $x^-$ in a dictionary. Denoting the query by $q = f(x)$, the correct key by $k^+ = h(x^+)$, and the dictionary of $K$ negative samples by $\{k_i = h(x_i)\}, i \in [1, K]$, we can express equation 1 in a softmax form, $\min_{\theta_q, \theta_k} \mathbb{E}_{x \backsim p_\mathcal{X}} \left[ -log \frac{e^{q \cdot k^+ / \tau}}{\sum_{i=0}^K e^{q \cdot k_i / \tau}} \right]$, where $\theta_q$ and $\theta_k$ are parameters of the query and key encoders, respectively, and $\tau$ is a temperature term that controls the shape of the probability distribution computed by the softmax function.

**Momentum Contrast (MoCo)** decouples the dictionary size from the mini-batch size by implementing a queue-based dictionary, i.e., current mini-batch samples are enqueued while the oldest are dequeued (He et al., 2020). It then applies momentum updates to parameters of a key encoder $\theta_k$ with respect to parameters of a query encoder, $\theta_k \leftarrow m\theta_k + (1 - m)\theta_q$, where $m \in [0, 1)$ is a momentum coefficient. Only the parameters $\theta_q$ are updated by back-propagation, while the parameters $\theta_k$ are defined as a moving average of $\theta_q$ with exponential smoothing. These two modifications allow MoCo to build a large and slowly-changing (and thus consistent) dictionary.

**Theoretical Limitations of Contrastive Learning.** Recent work provides theoretical analysis of the shortcomings of contrastive learning. McAllester & Stratos (2020) show that lower bounds to the MI are only tight for sample size exponential in the MI, suggesting that a large amount of data are required to achieve a tighter lower bound on MI. He et al. (2020) empirically showed that increasing negative samples has shown to improve the learned presentations. However, Arora et al. (2019) showed that such a phenomenon does not always hold: Excessive negative samples can sometimes hurt performance. Also, when the number of negative samples is large, the chance of sampling redundant instances increases, limiting the effectiveness of contrastive learning. One of our main contributions is to address this issue with active sampling of negative instances, which reduces redundancy and improves diversity, leading to improved performance on various downstream tasks.

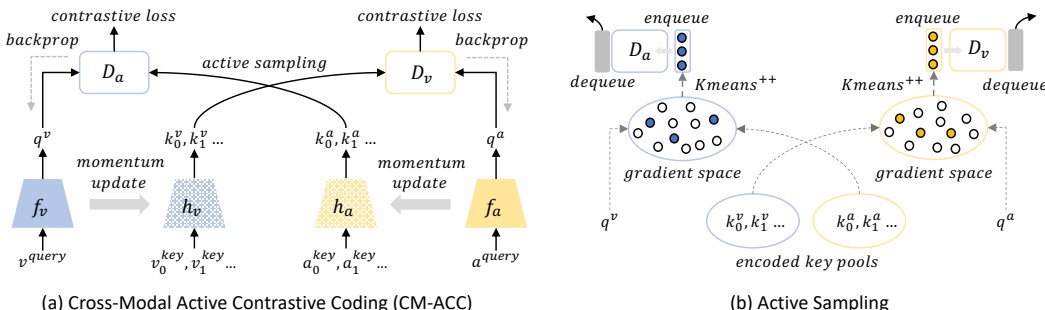

(a) Cross-Modal Active Contrastive Coding (CM-ACC)   (b) Active Sampling

Figure 1: **(a)** We extend contrastive learning to the cross-modal scenario and adapt momentum contrast (MoCo) (He et al., 2020) to the dictionary update. Different from all existing work, we propose an active learning idea to the negative sampling. **(b)** To sample negatives, we use the gradient space of our key encoders to estimate the *uncertainty* of each candidate in audio/visual pools, and take a *diverse* set of negatives in that space using the $k\text{-MEANS}_{\texttt{INIT}}^{++}$ algorithm.

## 3 APPROACH

### 3.1 CROSS-MODAL CONTRASTIVE REPRESENTATION LEARNING

Our learning objective encourages the representations of audio and visual clips to be similar if they come from the same temporal block of a video. Let $A = \{a_0, \cdots, a_{N-1}\}$ and $V = \{v_0, \cdots, v_{N-1}\}$ be collections of audio and visual clips, where each pair $(a_i, v_i)$ is from the same block of a video. We define query encoders $f_a$ and $f_v$ and key encoders $h_a$ and $h_v$ for audio and visual clips, respectively, with learnable parameters $\{\theta_q^a, \theta_q^v\}$ for the query encoders and $\{\theta_k^a, \theta_k^v\}$ for the key encoders. These encoders compute representations of audio and visual clips as queries and keys,

$$q^v = f_v(v^{query}), \quad k^v = h_v(v^{key}), \quad q^a = f_a(a^{query}), \quad k^a = h_a(a^{key}) \tag{2}$$

We train our encoders to perform cross-modal dictionary look-up, e.g., given a query video clip $v^{query}$, we find the corresponding audio clip $a^{key}$ from a dictionary $D_a$. Adapting MoCo (He et al., 2020) to our cross-modal setup, we implement a queue-based dictionary $D_a$ that stores keys of audio clips $\{k_i^a\}_{i=1}^K$, where $K$ is the dictionary size. We compute the contrastive loss and backpropagate the gradients only to the visual query encoder $f_v$ and update the parameters $\theta_q^v$. For the audio encoder $h_a$, we apply the momentum update (He et al., 2020),

$$\theta_k^a \leftarrow m\theta_k^a + (1 - m)\theta_q^a \tag{3}$$

The parameter $\theta_q^a$ is not updated in this contrastive coding step; we update it during the audio-to-visual step (similar as above with the opposite modalities). Here we explain the visual-to-audio step only; we perform bi-directional contrastive coding and train the whole model end-to-end.

### 3.2 ACTIVE SAMPLING OF NEGATIVE INSTANCES: UNCERTAINTY AND DIVERSITY

The quality of negative samples is crucial in contrastive learning. Existing work typically adopts random negative sampling. However, we want a diverse set of negative samples so that comparisons between positive and negative pairs are the most informative they can be. Motivated by active learning (Settles, 2009), we propose a gradient-based active sampling approach to improve the quality of negative samples. In active learning, the learner chooses samples that seem maximally informative and queries an oracle for labels to obtain an optimal solution with a minimal labeling budget. Adapting this to our setting, we can empower the learner to choose the maximally informative negative samples to construct a dictionary; the main question is how to measure the *informativeness* of samples without labels.

One way to measure informativeness is through the lens of *uncertainty*: If a model is highly uncertain about its prediction of a sample, we can ensure the maximum update to the model by including the sample in a mini-batch (conversely, if the uncertainly is low for all samples in a mini-batch, the model update will be small). Ash et al. (2020) showed that gradients of a loss function with respect to the model's most confident predictions can approximate the uncertainty of samples, demonstrating its effectiveness in active learning. They provide a theoretical justification by showing that gradient

norms of the last layer of a neural network with respect to pseudo-labels provides a lower bound on gradient norms induced by any other labels. In this work, we use gradients of the last layer to measure the uncertainty and encourage our model to include samples that have the highest gradient magnitudes to constitute a dictionary.

While the uncertainty of each individual samples is important, the *diversity* of samples is also a critical measure of informativeness. Intuitively, it is possible that a model is highly uncertain about samples from particular semantic categories, but constructing a mini-batch of samples from just those categories can severely bias gradients and ultimately lead to a bad local minima. There are several principled approaches to ensure diversity, e.g., submodular optimization (Fujishige, 2005) and Determinantal Point Processes (DPP) (Macchi, 1975; Kulesza & Taskar, 2011). Unfortunately, those methods are typically inefficient because of the combinatorial search space (Nemhauser et al., 1978; Gilks et al., 1995). In this work, instead of using the expensive solutions, we opt to the fast solution of Ash et al. (2020) and use the initialization scheme of the $k$-MEANS++ seeding algorithm (Arthur & Vassilvitskii, 2007) to sample a diverse set of negative samples.

## 3.3 CROSS-MODAL ACTIVE CONTRASTIVE CODING

Algorithm 1 describes our proposed cross-modal active contrastive coding (we provide a simplified version here; we include a more detailed version and another version without active sampling in Appendix). At a high-level, we initialize the dictionaries $D_v$ and $D_a$ with $K$ randomly drawn samples from $V$ and $A$, respectively (lines 3-4). For each epoch, we construct "negative candidate pools" $U_v$ and $U_a$ with $N$ random samples from $V$ and $A$, respectively (lines 6-7). For each iteration within an epoch, we *actively* select the most informative negative samples $S_v$ and $S_a$ from the pools $U_v$ and $U_a$, respectively, and enqueue them into the dictionaries $D_v$ and $D_a$, respectively (lines 9-21). We then perform cross-modal contrastive coding, update the parameters of query encoders $\theta_q^v$ and $\theta_q^a$ via backpropagation, and apply momentum updates to the parameters of key encoders $\theta_k^v$ and $\theta_k^a$ (lines 22-27).

---

**Algorithm 1** Cross-Modal Active Contrastive Coding

1: **Require:** Audio-visual clips $A, V$; encoders $f_v, f_a, h_v, h_a$; dictionary size $K$; pool size $N$; batch size $M$
2: Initialize parameters, $\theta_q^v, \theta_k^v, \theta_q^a, \theta_k^a \backsim Uniform(0,1)$
3: Draw random dictionary, $D_v \leftarrow \{v_1, \cdots, v_K\} \backsim V, D_a \leftarrow \{a_1, \cdots, a_K\} \backsim A$
4: Encode dictionary samples, $k_i^v \leftarrow h_v(v_i), \forall v_i \in D_v, k_i^a \leftarrow h_a(a_i), \forall a_i \in D_a$
5: **for** $epoch = 1$ **to** #epochs: **do**
6:     Draw random pool, $U_v \leftarrow \{v_1, \cdots, v_N\} \backsim V, U_a \leftarrow \{a_1, \cdots, a_N\} \backsim A$
7:     Encode pool samples, $k_n^v \leftarrow h_v(v_n), \forall v_n \in U_v, k_n^a \leftarrow h_a(a_n), \forall a_n \in U_a$
8:     **for** $t = 1$ **to** #mini-batches: **do**
9:         Draw mini-batch, $B_v \leftarrow \{v_1, \cdots, v_M\} \backsim V, B_a \leftarrow \{a_1, \cdots, a_M\} \backsim A$
10:          ▷ *Active sampling of negative video keys for $D_v$*
11:         Encode mini-batch samples, $q_i^a \leftarrow f_a(a_i), \forall a_i \in B_a$
12:         Compute pseudo-labels, $\tilde{y}_n^v \leftarrow \arg\max p(\hat{y}_n^v | v_n, B_a), \forall v_n \in U_v \backslash D_v$
13:         Compute gradients $g_{v_n}$ using the pseudo-labels $\tilde{y}_n^v, \forall n \in [1, N]$
14:         Obtain $S_v \leftarrow k\text{-MEANS}_{\text{INIT}}^{++}(\{g_{v_n} : v_n \in U_v \backslash D_v\}, \#\text{seeds} = M)$
15:         Update $D_v \leftarrow \text{ENQUEUE}(\text{DEQUEUE}(D_v), S_v)$
16:          ▷ *Active sampling of negative audio keys for $D_a$*
17:         Encode mini-batch samples, $q_i^v \leftarrow f_v(v_i), \forall v_i \in B_v$
18:         Compute pseudo-label, $\tilde{y}_n^a \leftarrow \arg\max p(\hat{y}_n^a | a_n, B_v), \forall a_n \in U_a \backslash D_a$
19:         Compute gradients $g_{a_n}$ using the pseudo-labels $\tilde{y}_n^a, \forall n \in [1, N]$
20:         Obtain $S_a \leftarrow k\text{-MEANS}_{\text{INIT}}^{++}(\{g_{a_n} : a_n \in U_a \backslash D_a\}, \#\text{seeds} = M)$
21:         Update $D_a \leftarrow \text{ENQUEUE}(\text{DEQUEUE}(D_a), S_a)$
22:          ▷ *Cross-modal contrastive predictive coding*
23:         Encode mini-batch samples, $k_i^v \leftarrow h_v(v_i), \forall v_i \in B_v, k_i^a \leftarrow h_a(a_i), \forall a_i \in B_a$
24:         Compute $p(y_i^v | v_i, a_i, D_a)$ and $p(y_i^a | a_i, v_i, D_v), \forall i \in [1, M]$
25:          ▷ *Update model parameters*
26:         Update parameters of query encoders $\theta_q^v$ and $\theta_q^a$ with backpropagation
27:         Momentum update parameters of key encoders $\theta_k^v$ and $\theta_k^a$
28:     **end for**
29: **end for**
30: **return** Optimal solution $\theta_q^v, \theta_k^v, \theta_q^a, \theta_k^a$

---

**Active sampling.** To measure *uncertainty*, we define a pseudo-label space induced by the queries from the other modality, and take the gradient of the last layer of a query encoder with respect to the most confident prediction, which we call the pseudo-label $\tilde{y}$. For instance, in the case of sampling negative video keys from the pool $U_v$ (`lines 10-15`), we compute the pseudo-posterior of a video key $v_n \in U_v \backslash D_a$,

$$p(\hat{y}_n^v | v_n, B_a) = \frac{\exp(k_n^v \cdot q_j^a)}{\sum_{i=1}^{M} \exp(k_n^v \cdot q_i^a)}, \forall j \in [1, M] \tag{4}$$

where $B_a$ is the current mini-batch of audio queries and defines the pseudo-label space. Note that we consider only the samples in $U_v \backslash D_v$ to rule out samples already in $D_v$. Intuitively, this computes the posterior by the dot-product similarity between $v_n$ and all $q_i^a \in B_a$, producing an $M$-dimensional probability distribution. We then take the most confident class category as the pseudo-label $\tilde{y}_n^v$ (`line 12`) and compute the gradient according to the cross-entropy loss

$$g_{v_n} = \frac{\partial}{\partial \theta_{last}} \mathcal{L}_{CE} \left( p(\hat{y}_n^v | v_n, B_a), \tilde{y}_n^v \right) |_{\theta = \theta_q^a} \tag{5}$$

where $\theta_{last}$ is the parameters of the last layer of $\theta$ (in this case, $\theta_q^a$ of the audio query encoder $h_a$). Intuitively, the gradient $g_{v_n}$ measures the amount of change – and thus, the *uncertainty* – $v_n$ will bring to the audio query encoder $h_a$.

One can interpret this as a form of *online hard negative mining*: The gradient is measured with respect to the most probable pseudo-label $\tilde{y}_n^v$ induced by the corresponding audio query $q_j^a$. When we compute the contrastive loss, the same audio query will be maximally confused by $v_n$ with its positive key $v^+$ per dot-product similarity, and $v_n$ in this case can serve as a hard negative sample.

Next, we obtain the most diverse and highly uncertain subset $S_v \subseteq U_v \backslash D_v$ using the initialization scheme of $k$-MEANS$^{++}$ (Arthur & Vassilvitskii, 2007) over the gradient embeddings $g_v$ (`line 14`). The $k$-MEANS$^{++}$ initialization scheme finds the seed cluster centroids by iteratively sampling points with a probability in proportion to their squared distances from the nearest centroid that has already been chosen (we provide the exact algorithm in the Appendix). Intuitively, this returns a *diverse* set of instances sampled in a greedy manner, each of which has a high degree of *uncertainty* measured as its squared distances from other instances that have already been chosen. Finally, we enqueue $S_v$ into $D_v$ and dequeue the oldest batch from $D_v$ (`line 15`). We repeat this process to sample negative audio keys (`lines 16-21`); this concludes the active sampling process for $D_v$ and $D_a$.

**Cross-modal contrastive coding.** Given the updated $D_v$ and $D_a$, we perform cross-modal contrastive coding. For visual-to-audio coding, we compute the posteriors of all video samples $v_i \in B_v$ with respect to the negative samples in the audio dictionary $D_a$,

$$p(y_i^v | v_i, a_i, D_a) = \frac{\exp(q_i^v \cdot k_i^a / \tau)}{\sum_{j=0}^{K} \exp(q_i^v \cdot k_j^a / \tau)}, \forall i \in [1, M] \tag{6}$$

where the posterior is defined over a cross-modal space with one positive and $K$ negative pairs (`line 24`). Next, we backpropagate gradients only to the query encoders $f_v$ and $f_a$ (`line 26`),

$$\theta_q^v \leftarrow \theta_q^v - \gamma \nabla_\theta \mathcal{L}_{CE}(p(y^v | \cdot), y_{gt}^v)|_{\theta=\theta_q^v}, \ \ \theta_q^a \leftarrow \theta_q^a - \gamma \nabla_\theta \mathcal{L}_{CE}(p(y^a | \cdot), y_{gt}^a)|_{\theta=\theta_q^a} \tag{7}$$

while applying momentum update to the parameters of the key encoders $h_v$ and $h_a$ (`line 27`),

$$\theta_k^v \leftarrow m\theta_k^v + (1-m)\theta_q^v, \ \ \theta_k^a \leftarrow m\theta_k^a + (1-m)\theta_q^a \tag{8}$$

The momentum update allows the dictionaries to change their states slowly, thus making them consistent across iterations. However, our cross-modal formulation can cause inconsistency in dictionary states because the gradient used to update query encoders are not directly used to update the corresponding key encoders. To improve stability, we let the gradients flow in a cross-modal fashion, updating part of $f_v$ and $h_a$ using the same gradient signal from the contrastive loss. We do this by adding one FC layer on top of all encoders and applying momentum update to their parameters. For example, we apply momentum update to the parameters of the FC layer on top of $h_a$ using the parameters of the FC layer from $f_v$. We omit this in Alg. 1 for clarity but show its importance in our ablation experiments (`XMoCo (w/o fcl)` in Table 1).

| # | Approach | Pretrain Obj. | UCF101 | HMDB51 | ESC50 | Gains |
|---|----------|---------------|--------|--------|-------|-------|
| ① | Scratch | - | 63.3 | 29.7 | 54.3 | |
| ② | Supervised | Supervised | 86.9 | 53.1 | 78.3 | |
| ③ | SMoCo | Uni. rand. | 70.7 | 35.2 | 69.0 | |
| ④ | XMoCo (w/o fcl) | Cross rand. | 72.9 (↑2.2) | 37.5 (↑2.3) | 70.9 (↑1.9) | $\Delta(④ - ③)$ |
| ⑤ | XMoCo | Cross rand. | 74.1 (↑1.2) | 38.7 (↑1.2) | 73.0 (↑2.1) | $\Delta(⑤ - ④)$ |
| ⑥ | CM-ACC (w/o fcl) | Cross active | 75.8 (↓1.4) | 39.1 (↓1.5) | 77.3 (↓1.9) | $\Delta(⑥ - ⑦)$ |
| ⑦ | CM-ACC | Cross active | 77.2 (↑3.1) | 40.6 (↑1.9) | 79.2 (↑6.2) | $\Delta(⑦ - ⑤)$ |

Table 1: Top-1 accuracy of unimodal vs. cross-modal pretraining on downstream tasks.

## 4 RELATED WORK

Self-supervised learning has been studied in vision, language, and audio domains. In the image domain, one popular idea is learning representations by maximizing the MI between different views of the same image (Belghazi et al., 2018; Hjelm et al., 2018; Tian et al., 2019; He et al., 2020). In the video domain, several approaches have exploited the spatio-temporal structure of video data to design efficient pretext tasks, e.g. by adopting ordering (Sermanet et al., 2017; Wang et al., 2019b), temporal consistency (Dwibedi et al., 2019), and spatio-temporal statistics (Xu et al., 2019; Wang et al., 2019a; Han et al., 2019). In the language domain, the transformer-based approaches trained with the masked language model (MLM) objective has been the most successful (Devlin et al., 2019; Liu et al., 2019; Yang et al., 2019). Riding on the success of BERT (Devlin et al., 2019), several concurrent approaches generalize it to learn visual-linguistic representations (Lu et al., 2019; Li et al., 2020; Su et al., 2019; Tan & Bansal, 2019; Li et al., 2019). CBT (Sun et al., 2019a) and VideoBERT (Sun et al., 2019b) made efforts on adapting BERT-style pretraining for video.

Besides vision and language signals, several approaches learn audio-visual representations in a self-supervised manner (Owens et al., 2016; Arandjelovic & Zisserman, 2017; Owens & Efros, 2018; Owens et al., 2016). Recently, audio-visual learning has been applied to enable interesting applications beyond recognition tasks, such as sound source localization/separation (Zhao et al., 2018; Arandjelovic & Zisserman, 2018; Gao et al., 2018; Gao & Grauman, 2019a;b; Ephrat et al., 2018; Gan et al., 2020; Zhao et al., 2019; Yang et al., 2020) and visual-to-sound generation (Hao et al., 2018; Zhou et al., 2018). The work of Owens & Efros (2018), Korbar et al. (2018), and Alwassel et al. (2019) are similar in spirit to our own, but our technical approach differs substantially in the use of active sampling and contrastive learning.

Hard negative mining is used in a variety of tasks, such as detection (Li et al., 2020), tracking (Nam & Han, 2016), and retrieval (Faghri et al., 2017; Pang et al., 2019), to improve the quality of prediction models by incorporating negative examples that are more difficult than randomly chosen ones. Several recent work have focused on finding informative negative samples for contrastive learning. Wu et al. (2020) show that the choice of negative samples is critical in contrastive learning and propose variational extension to InfoNCE with modified strategies for negative sampling. Iscen et al. (2018) propose hard examples mining for effective finetuning of pretrained networks. Cao et al. (2020) utilize negative sampling to reduce the computational cost. In the context of audio-visual self-supervised learning, Korbar et al. (2018) sample negatives under the assumption that the smaller the time gap is between audio and visual clips of the same video, the harder it is to differentiate them (and thus they are considered hard negatives). Our proposed approach does not make such an assumption and estimates the hardness of negatives by directly analyzing the magnitude of the gradients with respect to the contrastive learning objective.

## 5 EXPERIMENTS

**Experimental Setting.** We use 3D-ResNet18 (Hara et al., 2018) as our visual encoders ($f_v$ and $h_v$) in most of the experiments. We also use R(2+1)D-18 (Tran et al., 2018) to enable a fair comparison with previous work (see Table 4). For audio encoders ($f_a$ and $h_a$), we adapt ResNet-18 (He et al., 2016) to audio signals by replacing 2D convolution kernels with 1D kernels. We employ Batch Normalization (BN) (Ioffe & Szegedy, 2015) with the shuffling BN (He et al., 2020) in all our encoders. All models are trained end-to-end with the ADAM optimizer (Kingma & Ba, 2014) with an initial learning rate $\gamma = 10^{-3}$ after a warm-up period of 500 iterations. We use the mini-batch

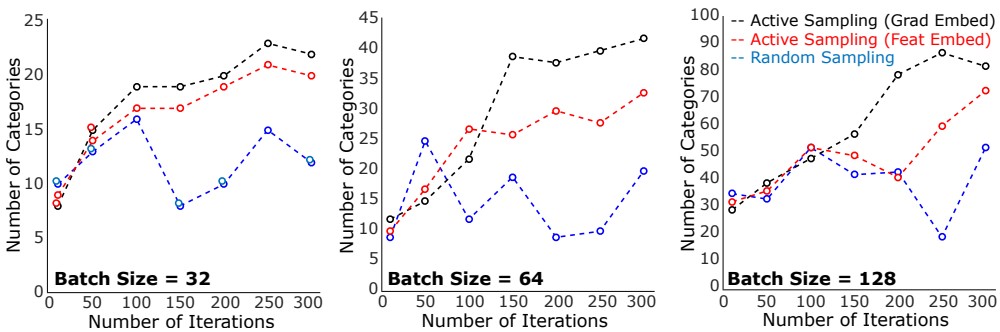

Figure 2: Effects of random sampling and active sampling on the number of categories.

size $M = 128$, dictionary size $K = 30 \times 128$, pool size $N = 300 \times 128$, momentum $m = 0.999$, and temperature $\tau = 0.7$. We used 40 NVIDIA Tesla P100 GPUs for our experiments.

We pretrain our model on Kinetics-700 (Carreira et al., 2019) and AudioSet (Gemmeke et al., 2017) when comparing with state-of-the-art approaches. For Kinetics-700, we use 240K randomly selected videos that contain the audio channel. On AudioSet, we use both a subset of 240K randomly selected videos and the 1.8M full set. For our ablation study, we use Kinetics-Sound (Arandjelovic & Zisserman, 2017) that contains 22K videos from 34 classes that are potentially manifested both visually and audibly, and thus provides a relatively clean testbed for ablation purposes. As for downstream tasks, we evaluate our models on action recognition using UCF101 (Soomro et al., 2012) and HMDB51 (Kuehne et al., 2011), and on sound classification using ESC50 (Piczak, 2015b).

**Unimodal vs. cross-modal pretraining.** To validate the benefits of cross-modal pretraining, we compare it to its unimodal counterparts. We pretrain our model on Kinetics-Sound with a randomly sampled dictionary (similar to MoCo (He et al., 2020)); we call this XMoCo. For the unimodal case, we pretrain two models on visual clips and audio clips, respectively; we call these SMoCo. We also compare ours with a model trained from scratch (Scratch), and a model pretrained on Kinetics-Sound in a fully-supervised manner (Supervised). Lastly, we include XMoCo (w/o fcl) that is identical to XMoCo except that we do not include the additional FC layers on top of the encoders. All these models are finetuned end-to-end on each downstream task using the same protocol.

Table 1 shows the top-1 accuracy of each downstream task. We observe that all the self-supervised models outperform Scratch on all downstream tasks, suggesting the effectiveness of pretraining with contrastive learning. We also see that our cross-modal objective outperforms the unimodal objective ($\Delta$(④-③)). The comparisons between XMoCo vs. XMoCo (w/o fcl) and CM-ACC vs. CM-ACC (w/o fcl) show the effectiveness of the additional FC layer on top of the encoders ($\Delta$(⑤-④), $\Delta$(⑥-⑦)). When adding the FC layer, the performance further improves on all three benchmarks. This shows the importance of letting the gradients flow in a cross-modal fashion. Finally, the performance gap with the full-supervised case shows there is still room for improvement in the self-supervised approaches.

Next, we compare the number of unique categories the sampled instances originally belong to, using the ground-truth labels provided in the dataset. Our logic is that the more categories the samples come from, the more diverse and less redundant the samples are. We train these on UCF-101 over 300 iterations with different mini-batch sizes, $M \in \{32, 64, 128\}$. Fig. 2 shows that active sampling selects more categories than random sampling across all three mini-batch sizes. At $M = 128$, active sampling (with gradient embedding) covers 60-70% of categories on UCF101, which is substantially more diverse than random sampling (30-40%). (A plot showing the probability of sampling unique negatives (instances from different categories) is shown in Appendix Figure 4.) While both sampling schemes perform similarly in early iterations, active sampling starts choosing more diverse instances as the training progresses; this is because the gradient embedding becomes more discriminative with respect to the uncertainty.

**Random vs. active sampling.** To validate the benefit of active sampling over random sampling, we compare models pretrained with different sampling approaches on downstream tasks. As shown in Table 1, our CM-ACC outperforms the XMoCo, which uses random sampling, by large margins, i.e. 3.1%, 1.9%, and 6.2% on UCF101, HMDB51, and ESC50, respectively ($\Delta$(⑦-⑤)).

| Pretrain Objective | Embedding Space | UCF101 | HMDB51 | ESC50 |
|---|---|---|---|---|
| Cross-modal active | Feature Embedding | 74.5 | 38.2 | 75.1 |
| Cross-modal active | Gradient Embedding | 77.2 (↑2.7) | 40.6 (↑2.4) | 79.2 (↑4.1) |

Table 2: Top-1 accuracy on downstream tasks: feature- vs. gradient-based embedding.

| | UCF101 | | | HMDB51 | | |
|---|---|---|---|---|---|---|
| | $M$=32 | $M$=64 | $M$=128 | $M$=32 | $M$=64 | $M$=128 |
| Random | 61.9 | 63.1 | 66.9 | 33.1 | 33.8 | 35.8 |
| OHEM | 50.2 (-11.7) | 60.8 (-2.3) | 65.7 (-1.2) | 26.8 (-6.3) | 30.1 (-3.7) | 33.2 (-2.6) |
| Active | 78.0 (+16.1) | 78.9 (+15.8) | 79.2 (+12.3) | 41.2 (+8.1) | 42.3 (+8.5) | 42.6 (+6.8) |

Table 3: Online hard example mining (OHEM) (Shrivastava et al., 2016) vs. our active sampling

**Feature vs. gradient embedding.** We compare two ways to do active sampling: using gradient embeddings (Eqn. 5) and feature embeddings (the outputs from $h_a$ and $h_v$) when selecting the seed centroids with $k$-MEANS$^{++}$. Fig. 2 shows that gradient embeddings produce a more diverse set of negative samples than feature embeddings; this is consistent across all three batch sizes. Table 2 shows that this diversity helps achieve better downstream performances across all three benchmarks. Fig. 5 (in Appendix) provides further insights, showing that the samples with high gradient magnitudes tend to be more informative negative samples.

From a theoretical aspect, the gradient norm induced by each candidate with computed pseudo labels estimates the candidates influence on the current model. The gradient embeddings convey information both about the model's uncertainty and potential update direction upon receiving a candidate. However, such messages are missing from the feature embeddings. This shows the importance of considering both uncertainty and diversity when selecting random samples: the $k$-MEANS$^{++}$ ensures the diversity in the sample set, but without the uncertainty measure we lose important discriminative information from the candidates.

**Online hard example mining vs. active sampling.** We compare our approach to online hard example mining (OHEM) (Shrivastava et al., 2016), which constructs negative samples by explicitly choosing the ones that incur high loss values. Specifically, we compute the pseudo-labels for all keys (negative sample candidates) with a given mini-batch of queries. We then compute the classification loss based on these pseudo labels and select the top $M$ keys with the highest loss values. We pretrain the models on Kinetics-700 (Kay et al., 2017) and report the top-1 accuracy on UCF101 (Soomro et al., 2012) and HMDB51 (Kuehne et al., 2011). We use the same architecture and hyperparameters; the only difference is the sampling approach.

Table 3 shows OHEM is generally less effective than both random sampling and our active sampling. Intuitively, OHEM promotes the dictionary to contain the most challenging keys for a given mini-batch of queries. Unfortunately, this causes OHEM to produce a *redundant* and *biased* dictionary, e.g., negative samples coming from a particular semantic category. Our results show that, when $M$ (mini-batch size) is small, the performance of OHEM is even worse than random sampling, although the gap between OHEM and random sampling decreases as $M$ increases. We believe this is because OHEM has a higher chance of selecting similar negative instances. When $M$ is large, this issue can be mitigated to some extent, but the performance still falls behind ours by a large margin. This suggests the importance of having a *diverse* set of negative samples, which is unique in our approach.

**Comparisons with SOTA.** Table 4 shows our approach outperforms various self-supervised approaches on action recognition. For fair comparisons, we group the SOTA approaches by different pretraining dataset sizes, i.e. small-scale (UCF/HMDB), medium-scale (Kinetics), and large-scale (AudioSet). Our gains are calculated according to this grouping. As we can see, our approach outperforms SOTA approaches across all groups. Compared with GDT (Patrick et al., 2020), the current top performing model on cross-modal self-supervised learning, our model outperforms it by 1.6 % on UCF101 and 1.1 % on HMDB51. Table 5 shows audio classification transfer results. For Kinetics and AudioSet (240K), our model outperforms the current state-of-the-art, AVID (79.1%) by 0.1% and 1.8% on Kinetics and AudioSet 240K, respectively. Our approach also outperforms AVID (89.2%) pretrained on AudioSet (1.8M) by 1.6%.

| Method | Architecture | Pretrained on (size) | UCF101 | HMDB51 |
|---|---|---|---|---|
| Scratch | 3D-ResNet18 | - | 46.5 | 17.1 |
| Supervised (Patrick et al., 2020) | R(2+1)D-18 | Kinetics400 (N/A) | 95.0 | 70.4 |
| ShufflAL (Misra et al., 2016) | CaffeNet | UCF/HMDB | 50.2 | 18.1 |
| DRL (Buchler et al., 2018) | CaffeNet | UCF/HMDB | 58.6 | 25.0 |
| OPN (Lee et al., 2017) | VGG | UCF/HMDB | 59.8 | 23.8 |
| DPC (Han et al., 2019) | 3D-ResNet18 | UCF101 | 60.6 | - |
| MotionPred (Wang et al., 2019a) | C3D | Kinetics400 (N/A) | 61.2 | 33.4 |
| RotNet3D (Jing & Tian, 2018) | 3D-ResNet18 | Kinetics400 (N/A) | 62.9 | 33.7 |
| ST-Puzzle (Kim et al., 2019) | 3D-ResNet18 | Kinetics400 (N/A) | 65.8 | 33.7 |
| ClipOrder (Xu et al., 2019) | R(2+1)D-18 | Kinetics400 (N/A) | 72.4 | 30.9 |
| CBT (Sun et al., 2019a) | S3D & BERT | Kinetics600 (500K) | 79.5 | 44.6 |
| DPC (Han et al., 2019) | 3D-ResNet34 | Kinetics400 (306K) | 75.7 | 35.7 |
| SeLaVi (Asano et al., 2020) | R(2+1)D-18 | Kinetics400 (240K) | 83.1 | 47.1 |
| AVTS (Korbar et al., 2018) | MC3 | Kinetics400 (240K) | 85.8 | 56.9 |
| XDC (Alwassel et al., 2019) | R(2+1)D-18 | Kinetics400 (240K) | 84.2 | 47.1 |
| AVID (Morgado et al., 2020) | R(2+1)D-18 | Kinetics400 (240K) | 87.5 | 60.8 |
| GDT (Patrick et al., 2020) | R(2+1)D-18 | Kinetics400 (N/A) | 89.3 | 60.0 |
| AVTS (Korbar et al., 2018) | MC3 | AudioSet (240K) | 86.4 | – |
| AVTS (Korbar et al., 2018) | MC3 | AudioSet (1.8M) | 89.0 | 61.6 |
| XDC (Alwassel et al., 2019) | R(2+1)D-18 | AudioSet (1.8M) | 91.2 | 61.0 |
| AVID (Morgado et al., 2020) | R(2+1)D-18 | AudioSet (1.8M) | 91.5 | 64.7 |
| GDT (Patrick et al., 2020) | R(2+1)D-18 | AudioSet (1.8M) | 92.5 | 66.1 |
| | 3D-ResNet18 | UCF101 | 69.1 (+8.5) | 33.3 (+8.3) |
| | 3D-ResNet18 | Kine.-Sound (14K) | 77.2 (+16.6) | 40.6 (+15.6) |
| Ours | 3D-ResNet18 | Kinetics700 (240K) | 90.2 (+0.9) | 61.8 (+1.0) |
| | 3D-ResNet18 | AudioSet (240K) | 90.7 (+1.4) | 62.3 (+1.5) |
| | 3D-ResNet18 | AudioSet (1.8M) | **94.1** (+1.6) | 66.8 (+0.7) |
| | R(2+1)D-18 | AudioSet (1.8M) | 93.5 (+1.0) | **67.2** (+1.1) |

Table 4: Comparison of SOTA approaches on action recognition. We specify pretraining dataset and the number of samples used if they are reported in the original papers (N/A: not available).

| Method | Architecture | Pretrained on (size) | ESC50 |
|---|---|---|---|
| Random Forest (Piczak, 2015b) | MLP | ESC50 | 44.3 |
| Piczak ConvNet (Piczak, 2015a) | ConvNet-4 | ESC50 | 64.5 |
| ConvRBM (Sailor et al., 2017) | ConvNet-4 | ESC50 | 86.5 |
| SoundNet (Aytar et al., 2016) | ConvNet-8 | SoundNet (2M+) | 74.2 |
| $L^3$-Net (Arandjelovic & Zisserman, 2017) | ConvNet-8 | SoundNet (500K) | 79.3 |
| AVTS (Korbar et al., 2018) | VGG-8 | Kinetics (240K) | 76.7 |
| XDC (Alwassel et al., 2019) | ResNet-18 | Kinetics (240K) | 78.0 |
| AVID (Morgado et al., 2020) | ConvNet-9 | Kinetics (240K) | 79.1 |
| AVTS (Korbar et al., 2018) | VGG-8 | AudioSet (1.8M) | 80.6 |
| XDC (Alwassel et al., 2019) | ResNet-18 | AudioSet (1.8M) | 84.8 |
| AVID (Morgado et al., 2020) | ConvNet-9 | AudioSet (1.8M) | 89.2 |
| GDT (Patrick et al., 2020) | ResNet-9 | AudioSet (1.8M) | 88.5 |
| | | Kinetics700 (240K) | 80.2 (+1.1) |
| Ours | ResNet-18 | AudioSet (240K) | 80.9 (+1.8) |
| | | AudioSet (1.8M) | **90.8** (+1.6) |

Table 5: Comparision of SOTA approaches on audio event classification.

# 6 CONCLUSION

We have shown that random sampling could be detrimental to contrastive learning due to the redundancy in negative samples, especially when the sample size is large, and have proposed an active sampling approach that yields diverse and informative negative samples. We demonstrated this on learning audio-visual representations from unlabeled videos. When pretrained on AudioSet, our approach outperforms previous state-of-the-art self-supervised approaches on various audio and visual downstream benchmarks. We also show that our active sampling approach significantly improves the performance of contrastive learning over random and online hard negative sampling approaches.

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

## A  DETAILS ON DATA PROCESSING

We preprocess video frames by sampling at 10 FPS and applying random cropping, horizontal flipping, gray-scaling, and temporal jittering. We resize video frames to 3-channel images of $224 \times 224$; we set the clip length to 16 frames during pretraining, and 32 frames during finetuning on downstream tasks. For audio channel, we extract mel-spectrograms from the raw waveform using the LibROSA library and get a $80 \times T$ matrix with 80 frequency bands; T is proportionate to the length of an audio clip. We then segment the mel-spectrogram according to the corresponding video clips to ensure temporal synchrony. We treat the mel-spectrograms as an 80-channel 1D signal.

As for downstream tasks, we evaluate our models on action recognition using UCF101 (Soomro et al., 2012) and HMDB51 (Kuehne et al., 2011), and on sound classification using ESC50 (Piczak, 2015b). UCF101 contains 13K video clips from 101 action categories, HMDB51 contains 7K video clips from 51 categories, and ESC50 has 2K audio clips from 50 categories. UCF101 and HMDB51 have 3 official train/test splits, while ESC50 has 5 splits. We conduct our ablation study using `split-1` of each dataset. We report our average performance over all splits when we compare with prior work.

## B  ADDITIONAL EXPERIMENTS

**Effect of mutual information.** We investigate the impact of the amount of MI on contrastive learning using the Spatial-MultiOmniglot dataset (Ozair et al., 2019). It contains paired images $(x, y)$ of Omniglot characters (Lake et al., 2015) with each image arranged in an $m \times n$ grid (each grid cell is $32 \times 32$ pixels). Let $l_i$ be the alphabet size for the $i^{th}$ character in each image, then the MI $I(x, y) = \sum_{i=1}^{mn} log l_i$. This way, we can easily control the MI by adding or removing characters. We follow the experimental protocol of Ozair et al. (Ozair et al., 2019), keeping the training dataset size fixed at 50K and using the same alphabet sets: Tifinagh (55 characters), Hiragana (52), Gujarati (48), Katakana (47), Bengali (46), Grantha (43), Sanskrit (42), Armenian (41), and Mkhedruli (41).

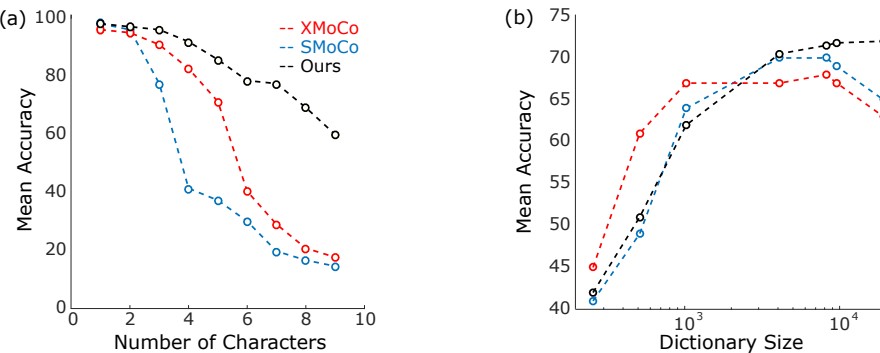

Figure 3: The effect of a) mutual information (Spatial-MultiOmniglot) and b) dictionary size on the accuracy of classification (UCF101).

Fig. 3(a) shows the results as the number of characters (and thus the MI) increases. We see that all approaches achieve nearly 99% accuracy with less than 3 characters; this is the case when the exponent of the MI is smaller than the dataset size (50K), i.e., $e^{I(x,y)} = 55$ with one character, $e^{I(x,y)} = 2,860$ with 2 characters. However, starting from 3 characters, the performance of the regular MoCo (SMoCo) drops significantly; this is because the exponent of the MI (=137,280 (55×52×48)) is much larger than the dataset size. Although our model also drops performance when the MI is increased, it outperforms the other approaches by a large margin. We also observe that XMoCo outperforms SMoCo in mild conditions (1-5 characters) but performs nearly the same as SMoCo with severe conditions (6-9 characters). This suggests that, while cross-modal prediction helps to learn good representations, it also suffers with the same issue when the MI is large, thus adopting active sampling is beneficial.

**Effect of dictionary size.** Fig. 3 (b) shows how the dictionary size affects downstream task performance. Here we pretrain our model on Kinetics-700 and finetune it on UCF-101. Overall, all

three approaches benefit from large dictionaries up to a threshold (at about $10^3$), which is consistent with previous empirical findings (He et al., 2020). However, both XMoCo and SMoCo starts deteriorating performance after about $10^4$ (which is consistent with previous theoretical claims of Arora et. al (Arora et al., 2019)), whereas ours do not suffer even after $10^4$. This suggests that there are performance limits by simply increasing the size of a randomly-sampled dictionary, and also shows the benefit of our active sampling approach.

**Effect of pretraining dataset sizes.** We investigate the effects of the size of pretraining datasets, using Kinetics-Sound (22k), Kinetics (240K), and AudioSet (1.8M). We vary pretraining conditions while using the same protocol to finetune the models end-to-end on downstream tasks.

Table 6 shows that our model benefits from pretraining on video data, and that the performance improves as we use a large pretraining video dataset (Kinetics and AudioSet) than the relatively smaller dataset (Kinetics-Sound). Notably, our approach even outperforms the fully-supervised pretraining approaches by pretraining on a larger video dataset (1.0%, 3.6%, and 8.5% improvement on UCF101, HMDB51, and ESC50, respectively.)

| Approach | Dataset | UCF101 | HMDB51 | ESC50 |
|---|---|---|---|---|
| Supervised | ImageNet (1.2M) | $82.8^\dagger$ | $46.7^\dagger$ | – |
|  | Kinetics-Sound (22K) | $86.9^*$ | $53.1^*$ | $78.3^*$ |
|  | Kinetics400 (240K) | $93.1^\dagger$ | $63.6^\dagger$ | $82.3^*$ |
| CM-ACC | Kinetics-Sound (22K) | 77.2 | 40.6 | 77.3 |
|  | Kinetics700 (240K) | 90.2 (-2.9) | 61.8 (-1.8) | 79.2 (-3.1) |
|  | AudioSet (1.8M) | 94.1 (+1.0) | 67.2 (+3.6) | 90.8 (+8.5) |

Table 6: Top-1 accuracy of CM-ACC pretrained on different datasets vs. fully-supervised counterparts (Supervised). $\dagger$: the results are excerpted from Patrick et al. (2020), $*$: our results.

**Diversity of random vs. active sampling.** To compare the diversity of the chosen negatives by random vs. active sampling, we plot the probability of them on sampling of unique negatives (instances from different categories). The more categories the samples come from, we get more diverse and less redundant samples. We train these on UCF-101 over 300 iterations with different mini-batch sizes, $M \in \{32, 64, 128\}$. As shown in Figure 4, the active sampling selects more categories than random sampling across all three mini-batch sizes. At $M = 128$, active sampling (with gradient embedding) covers 60-70% of categories on UCF101, which is substantially more diverse than random sampling (30-40%).

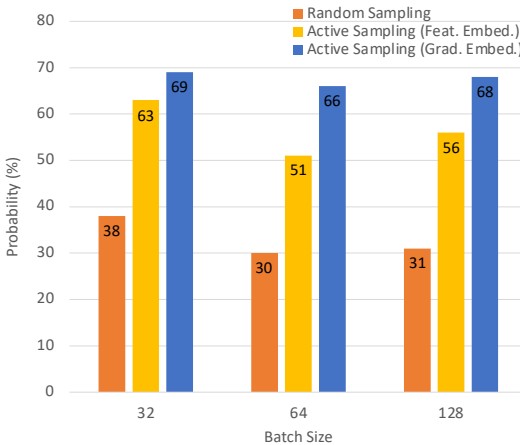

Figure 4: Probability of sampling unique negatives (instances from different categories) in the random vs. active sampling conditions. We compute the probabilities by averaging the number of unique categories across iterations and dividing them by their batch size.

## C VISUALIZATION OF NEGATIVE INSTANCES

Figure 5 shows negative instances selected by active sampling and random sampling when we use audio clips as the query. We visualize the center frames of the selected video clips. We can see that our approach selects more **challenging** examples than the random sampling approach. For instance,

given a query `opening bottle`, our approach selected video clips from the same or similar semantic categories, e.g. `drinking shot` and `opening bottle`. Given `snowboarding`, our approach selected more video clips related to categories containing the snow scene, e.g. `ice fishing`, `snow kiting`, and `tobogganing`.

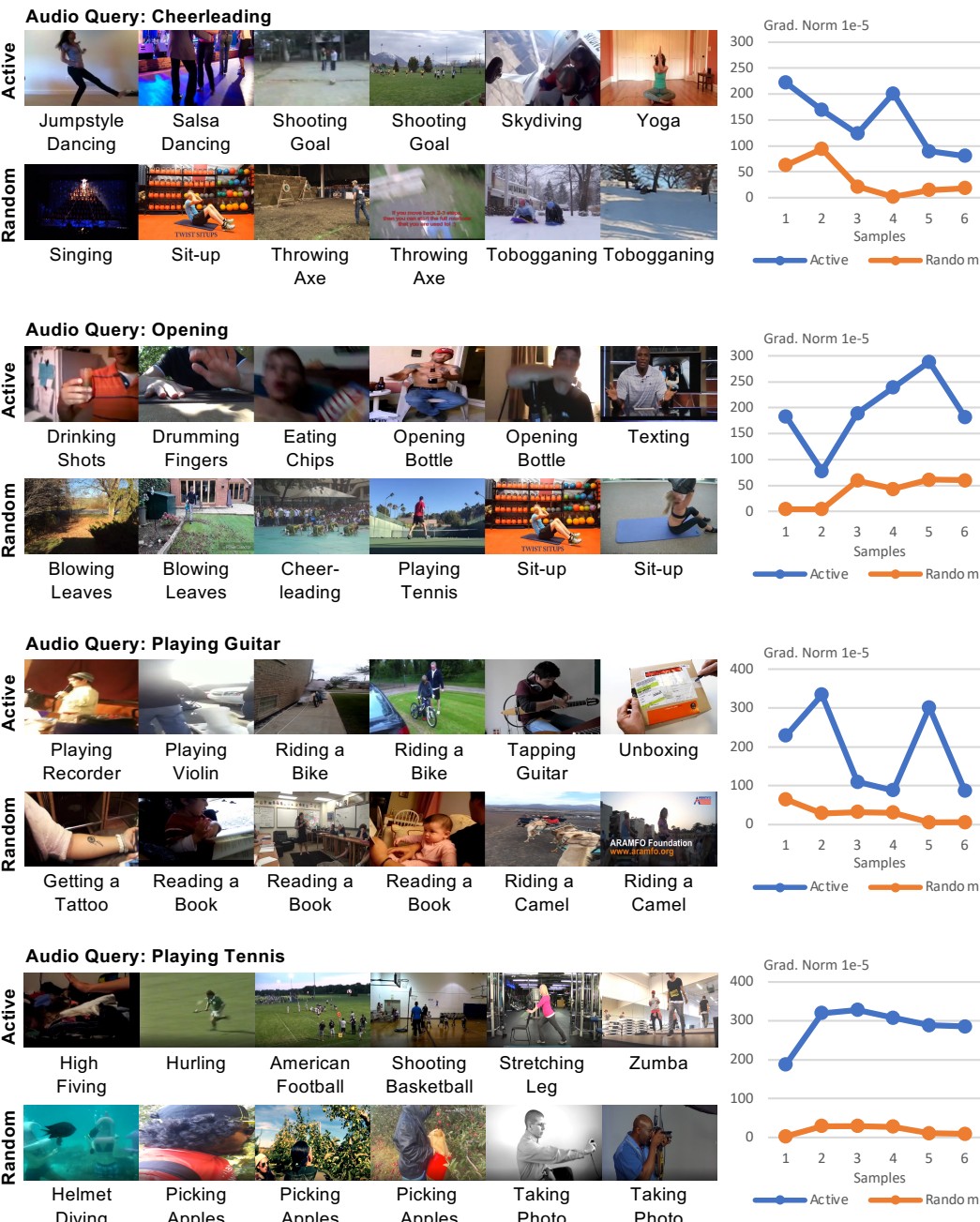

Figure 5: Center frames of video clips and their gradient norms selected by active sampling and random sampling.

Furthermore, we also find that our approach selects more *diverse* negative samples. For example, given a query `snowboarding`, active sampling selected video clips from 4 different categories related to the snow scene: (`ice fishing`, `playing ice hockey`, `snow kiting`, and `tobogganing`). In comparison, the random sampling approach yields fewer semantic cate-

gories in general. This suggests that our active sampling approach produces more 'challenging' and 'diverse' negative instances than the random sampling approach.

To clearly investigate the relationship between negative samples and their gradient magnitudes, we show the gradient norm of each visualized sample in Figure 5. We can see that hard negatives tend to have larger gradient norms than easy negatives. Given a query `Playing guitar`, video clips containing the concept of "playing instruments" yield the higher gradient norms, i.e. `playing violin` (333.87) and `tapping guitar` (301.35), while concepts that are easy to discriminate, e.g., `riding a camel` yield a significantly smaller gradient norm (5.92). This provides evidence showing the gradient magnitude is effective in measuring the uncertainty of the current model, i.e., highly-uncertain samples (hard negatives) tend to yield gradients with larger magnitudes, while highly-confident samples (easy negatives) tend to have smaller gradient magnitudes.

## D    WHEN WOULD CROSS-MODAL CONTRASTIVE LEARNING FAIL?

In general, cross-modal video representation learning is based on an assumption that the natural correspondence between audio and visual channels could serve as a useful source of supervision. While intuitive, this assumption may not hold for certain videos in-the-wild, which may cause the model to learn suboptimal representations. To investigate when our approach succeeds and fails, we conduct a post-hoc analysis by using thehttps://www.overleaf.com/project/5ded2abe1c17bc00011e5da8 ground-truth semantic category labels provided in Kinetics-700 (Carreira et al., 2019) (which is not used during pretraining). Specifically, we use our pretrained model to solve the audio-visual contrastive pretext task (Eqn.(7) in the main paper) and keep track of the prediction results (correct/incorrect). We then average the pretext task accuracy over 100 randomly chosen samples for each action category.

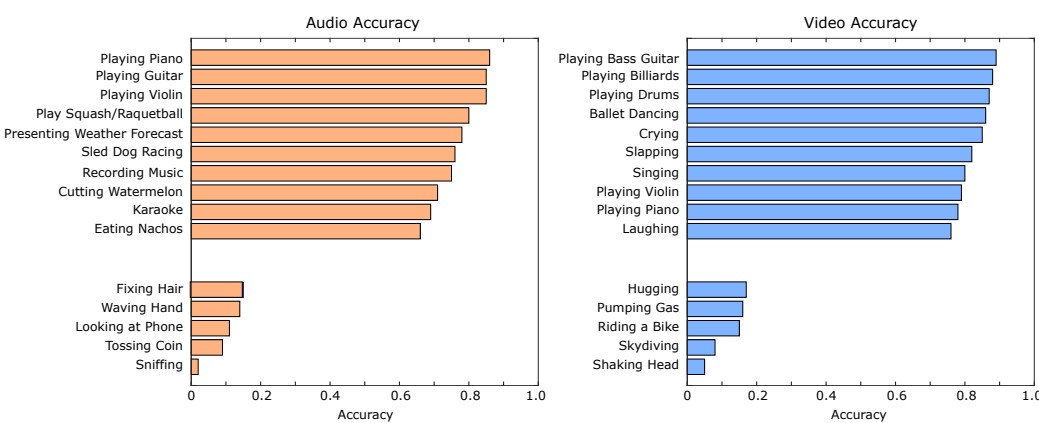

Figure 6: Distribution of Kinetics-700 (Carreira et al., 2019) categories sorted by the prediction accuracy.

Figure 6 shows the top-10 and bottom-5 classes by using both audio (left) and video (right) as the query. We observe that, the top ranked classes for both audio and video are the activities that have highly correlated visual-audio signals. For instance, `playing bass guitar`, `play piano`, and `play violin` are all activities related to music. The correlation of audio-visual signals for these activities are obvious; such highly correlated signals are easier to be learned in a cross-modal manner. On the contrary, the bottom ranked classes are those that have subtle audio-visual correlation, e.g. `tossing coin`, `shaking hand`, `looking at phone`, and `hugging`. We also investigate the distribution of hard-easy classes with that reported in Kinetics-700 (Carreira et al., 2019) learned by the I3D-RGB model (Carreira & Zisserman, 2017). Interestingly, we find that some hard classes (e.g. `karaoke` and `recording music`) are listed in our top ranked classes. We suspect that, when only learned within visual modality, some classes with cluttered or complected spatial information will bring difficulties for classification. While, as our cross-modal approach can

leverage information from both auditory and visual information, so our model does not limited by such problems.

---

**Algorithm 2** Cross-Modal Active Contrastive Coding (Detailed version of Algorithm 1)

1: **Require:** Audio-visual clips $A, V$; encoders $f_v, f_a, h_v, h_a$; dictionary size $K$; pool size $N$; batch size $M$
2: Initialize parameters, $\theta_q^v, \theta_k^v, \theta_q^a, \theta_k^a \backsim Uniform(0,1)$
3: Draw random dictionary, $D_v \leftarrow \{v_1, \cdots, v_K\} \backsim Random(V), D_a \leftarrow \{a_1, \cdots, a_K\} \backsim Random(A)$
4: Encode dictionary samples, $k_i^v \leftarrow h_v(v_i), \forall v_i \in D_v, k_i^a \leftarrow h_a(a_i), \forall a_i \in D_a$
5: **for** $epoch = 1$ **to** #epochs: **do**
6: $\quad$ Draw random pool, $U_v \leftarrow \{v_1, \cdots, v_N\} \backsim Random(V), U_a \leftarrow \{a_1, \cdots, a_N\} \backsim Random(A)$
7: $\quad$ Encode pool samples, $k_n^v \leftarrow h_v(v_n), \forall v_n \in U_v, k_n^a \leftarrow h_a(a_n), \forall a_n \in U_a$
8: $\quad$ **for** $t = 1$ **to** #mini-batches: **do**
9: $\quad\quad$ Draw mini-batch, $B_v \leftarrow \{v_1, \cdots, v_M\} \backsim V, B_a \leftarrow \{a_1, \cdots, a_M\} \backsim A$
10: $\quad\quad$ ▷ *Active sampling of negative video keys for $D_v$*
11: $\quad\quad$ Encode mini-batch samples, $q_i^a \leftarrow f_a(a_i), \forall a_i \in B_a$
12: $\quad\quad$ **for** $\forall v_n \in U_v \backslash D_v$: **do**
13: $\quad\quad\quad$ Compute pseudo-posterior, $p(\hat{y}_n^v | v_n, B_a) \leftarrow \frac{\exp(k_n^v \cdot q_j^a)}{\sum_{i=1}^M \exp(k_n^v \cdot q_i^a)}, \forall j \in [1, M]$
14: $\quad\quad\quad$ Compute pseudo-label, $\tilde{y}_n^v \leftarrow \arg\max p(\hat{y}_n^v | \cdot)$
15: $\quad\quad$ **end for**
16: $\quad\quad$ Compute gradient, $g_{v_n} \leftarrow \frac{\partial}{\partial \theta_{last}} \mathcal{L}_{CE}(p(\hat{y}_n^v | \cdot), \tilde{y}_n^v)|_{\theta=\theta_q^a}, \forall n \in [1, N]$
17: $\quad\quad$ Obtain $S_v \leftarrow k\text{-MEANS}_{\texttt{INIT}}^{++}(\{g_{v_n} : v_n \in U_v \backslash D_v\}, \#\text{seeds} = M)$
18: $\quad\quad$ Update $D_v \leftarrow \text{ENQUEUE}(\text{DEQUEUE}(D_v), S_v)$
19: $\quad\quad$ ▷ *Active sampling of negative audio keys for $D_a$*
20: $\quad\quad$ Encode mini-batch samples, $q_i^v \leftarrow f_v(v_i), \forall v_i \in B_v$
21: $\quad\quad$ **for** $\forall a_n \in U_a \backslash D_a$: **do**
22: $\quad\quad\quad$ Compute pseudo-posterior, $p(\hat{y}_n^a | a_n, B_v) \leftarrow \frac{\exp(k_n^a \cdot q_j^v)}{\sum_{i=1}^M \exp(k_n^a \cdot q_i^v)}, \forall j \in [1, M]$
23: $\quad\quad\quad$ Compute pseudo-label, $\tilde{y}_n^a \leftarrow \arg\max p(\hat{y}_n^a | \cdot)$
24: $\quad\quad$ **end for**
25: $\quad\quad$ Compute gradient, $g_{a_n} \leftarrow \frac{\partial}{\partial \theta_{last}} \mathcal{L}_{CE}(p(\hat{y}_n^a | \cdot), \tilde{y}_n^a)|_{\theta=\theta_q^v}, \forall n \in [1, N]$
26: $\quad\quad$ Obtain $S_a \leftarrow k\text{-MEANS}_{\texttt{INIT}}^{++}(\{g_{a_n} : a_n \in U_a \backslash D_a\}, \#\text{seeds} = M)$
27: $\quad\quad$ Update $D_a \leftarrow \text{ENQUEUE}(\text{DEQUEUE}(D_a), S_a)$
28: $\quad\quad$ ▷ *Cross-modal contrastive predictive coding*
29: $\quad\quad$ Encode mini-batch samples, $k_i^v \leftarrow h_v(v_i), \forall v_i \in B_v, k_i^a \leftarrow h_a(a_i), \forall a_i \in B_a$
30: $\quad\quad$ Compute $p(y_i^v | \cdot) = \frac{\exp(q_i^v \cdot k_i^a / \tau)}{\sum_{j=0}^K \exp(q_i^v \cdot k_j^a / \tau)}, p(y_i^a | \cdot) = \frac{\exp(q_i^a \cdot k_i^v / \tau)}{\sum_{j=0}^K \exp(q_i^a \cdot k_j^v / \tau)}, \forall i \in [1, M]$
31: $\quad\quad$ ▷ *Update model parameters*
32: $\quad\quad$ Update $\theta_q^v \leftarrow \theta_q^v - \gamma \nabla_\theta \mathcal{L}_{CE}(p(y^v | \cdot), y_{gt}^v)|_{\theta=\theta_q^v}, \theta_q^a \leftarrow \theta_q^a - \gamma \nabla_\theta \mathcal{L}_{CE}(p(y^a | \cdot), y_{gt}^a)|_{\theta=\theta_q^a}$
33: $\quad\quad$ Momentum update $\theta_k^v \leftarrow m\theta_k^v + (1-m)\theta_q^v, \theta_k^a \leftarrow m\theta_k^a + (1-m)\theta_q^a$
34: $\quad$ **end for**
35: **end for**
36: **return** Optimal solution $\theta_q^v, \theta_k^v, \theta_q^a, \theta_k^a$

---

**Algorithm 3** $k\text{-MEANS}_{\texttt{INIT}}^{++}$ Seed Cluster Initialization

1: **Require:** Data $X$ of $N$ samples; number of centroids $K$
2: Choose one centroid uniformly at random, $C[0] \leftarrow x \backsim Random(X)$
3: **for** $k = 1$ **to** $K - 1$: **do**
4: $\quad$ ▷ *Compute a cumulative probability distribution with a probability in proportion to their squared distances from the nearest centroid that has already been chosen*
5: $\quad$ **for** $n = 0$ **to** $N - 1$: **do**
6: $\quad\quad$ Compute the squared distance, $D[n] \leftarrow (min\_dist(X[n], C))^2$
7: $\quad$ **end for**
8: $\quad$ Compute the cumulative probability distribution, $P \leftarrow \frac{cumsum(D)}{sum(D)}$
9: $\quad$ ▷ *The next centroid is chosen using $P(X)$ as a weighted probability distribution*
10: $\quad$ Choose one centroid at random, $C[k] \leftarrow x \backsim P(X)$
11: **end for**
12: **return** $C$ containing $K$ centroids

---

**Algorithm 4** Cross-Modal Contrastive Coding *without* Active Sampling

---

1: **Require:** Audio-visual clips $A, V$; dictionary $D_v$; encoders $f_v, f_a, h_v, h_a$;
   dictionary size $K$; mini-batch size $M$; learning rate $\gamma$; momentum $m$
2: Initialize parameters, $\theta_q^v, \theta_k^v, \theta_q^a, \theta_k^a \backsim Uniform(0, 1)$
3: Load a dictionary at random, $D_a \leftarrow \{v_1, \cdots, v_K\} \backsim Random(V)$
4: Load a dictionary at random, $D_v \leftarrow \{a_1, \cdots, a_K\} \backsim Random(A)$
5: Encode dictionary samples, $k_i^v \leftarrow h_v(v_i), \forall v_i \in D_a, k_i^a \leftarrow h_a(a_i), \forall a_i \in D_v$
6: **for** $epoch = 1$ **to** #epochs: **do**
7:    **for** $t = 1$ **to** #mini-batches: **do**
8:       Load a mini-batch of visual clips, $B_v \leftarrow \{v_1, \cdots, v_M\} \backsim V$
9:       Load a mini-batch of audio clips, $B_a \leftarrow \{a_1, \cdots, a_M\} \backsim A$
10:       ▷ *Update dictionaries*
11:       Encode mini-batch samples, $k_i^v \leftarrow h_v(v_i), \forall v_i \in B_v$
12:       Encode mini-batch samples, $k_i^a \leftarrow h_a(a_i), \forall a_i \in B_a$
13:       Update $D_v \leftarrow \text{ENQUEUE}(\text{DEQUEUE}(D_v), B_v)$
14:       Update $D_a \leftarrow \text{ENQUEUE}(\text{DEQUEUE}(D_a), B_a)$
15:       ▷ *Cross-modal contrastive predictive coding*
16:       Encode mini-batch samples, $q_i^v \leftarrow f_v(v_i), \forall v_i \in B_v$
17:       Encode mini-batch samples, $q_i^a \leftarrow f_a(a_i), \forall a_i \in B_a$
18:       Compute the posterior, $p(y_i^v|v_i, a_i, D_v) = \frac{\exp(q_i^v \cdot k_i^a / \tau)}{\sum_{j=0}^K \exp(q_i^v \cdot k_j^a / \tau)}, \forall i \in [1, M]$
19:       Compute the posterior, $p(y_i^a|a_i, v_i, D_v) = \frac{\exp(q_i^a \cdot k_i^v / \tau)}{\sum_{j=0}^K \exp(q_i^a \cdot k_j^v / \tau)}, \forall i \in [1, M]$
20:       ▷ *Update model parameters*
21:       Update $\theta_q^v \leftarrow \theta_q^v - \gamma \nabla_\theta \mathcal{L}_{CE}(-\log p(y^v|\cdot), y_{gt}^v)|_{\theta=\theta_q^v}$
22:       Update $\theta_q^a \leftarrow \theta_q^a - \gamma \nabla_\theta \mathcal{L}_{CE}(-\log p(y^a|\cdot), y_{gt}^a)|_{\theta=\theta_q^a}$
23:       Momentum update $\theta_k^v \leftarrow m\theta_k^v + (1-m)\theta_q^v$
24:       Momentum update $\theta_k^a \leftarrow m\theta_k^a + (1-m)\theta_q^a$
25:    **end for**
26: **end for**
27: **return** Optimal solution $\theta_q^v, \theta_k^v, \theta_q^a, \theta_k^a$

---

