# OpenReview forum: "Active Contrastive Learning of Audio-Visual Video Representations"
_ICLR.cc/2021/Conference — ICLR 2021 Poster_

### Official Review · AnonReviewer2 · 2020-10-27
**Interesting multi-modal contrastive learning method, but needs some more empirical validation and fair comparison**

**Rating:** 7
**Confidence:** 3

**Review:**

#### Summary
In this paper, the authors propose a cross-modal (audio-video) self-supervised representation learning method with a contrastive learning framework. To overcome the high redundancy in the negative samples, they propose an active negative sampling method. They use a gradient with respect to the pseudo label to measure the uncertainty of a negative sample. They use K-means clustering to maximize the negative sample diversity when constructing a new negative set for queueing. They show their method's efficacy on the public benchmarks: Kinetics, AudioSet for retraining, and UCF-101, HMDB-51, ESC-50 for downstream tasks.

#### Strengths
* The problem is interesting and relevant to the community. Audio-visual self-supervised learning could be interesting for many researchers in the computer vision community.
* Design choice looks reasonable and effective. Cross-modal contrastive learning gives a significant boost to performance. Active sampling gives a significant boost.
* Writing is straightforward and easy to follow.

#### Weaknesses
However, I have a few concerns about this work.
* To show the effectiveness of the proposed gradient-based active sampling method compared to the embedding-based active sampling, the authors plot the number of action categories vs. the number of training iterations in Figure 2. However, in my opinion, more action categories might not mean high diversity in a negative set. For example, an action could be compositional. "Drinking coffee" action consists of "picking up object", "approach an object", "bring an object to the actor". Since the proposed method uses 16 frames, very short, feature encoding, the embeddings will be more towards these subactions, not the entire actions. Therefore, more action categories do not directly mean more diverse negative examples. So I would suggest the authors directly compare the downstream task performance of gradient-based sampling and embedding-based sampling.
* The comparison of the proposed method with the fully supervised method is unfair. The fully supervised method has VGG-M-2048 as a backbone and pretrained on ImageNet, while the proposed methods have 3D-ResNet-18 or R(2+1)D-18 as a backbone and pertained on Kinetics or AudioSet. It is also unclear which row of "ours" the authors compare to the fully supervised method. I cannot find a 7.2p improvement on the UCF-101 and the 14.1p improvement on the HMDB-51 compared to the fully supervised method. It is okay that the proposed self-supervised method is weaker than the fully supervised method. However, the comparison should be fair, in my opinion.

I will increase my rating if the authors adequately address my concerns.

#### Minor comments:
* 6. Conclusion, fourth row: "When pertained …" -> "When pre-trained …"?

---

> ### Author Response · Authors · 2020-11-21
> **Thank you! We added more empirical validation under fair settings.**
>
> Thank you for your positive feedback. We have addressed both of your comments in the revision, as summarized below:
>
> * RE: gradient- vs. feature-based embedding. We added new results (Table 2) to clearly demonstrate the efficacy of our gradient-based embedding over a feature-based embedding. We find that our approach outperforms the feature-based approach by a large margin across all three downstream benchmarks. This is because the gradient embeddings directly estimate the samples’ influence on the current model, which conveys information about the model’s uncertainty; this is missing in the feature embeddings. We add more detailed analysis in main paper (Sec. Feature vs. gradient embedding.).
>
> * RE: Comparison with a fully-supervised method. We removed this claim from our paper and updated Table 4 with fully-supervised results from GDT [Patrick et al., 2020], which show the highest accuracy compared to all the other self-supervised results using the same backbone (90.5% UCF101; 70.4% HMDB51). To clarify, the previous results that we included in our original submission (82.8% UCF101; 46.7% HMDB51) are from an older version of the XDC paper (https://arxiv.org/pdf/1911.12667v1.pdf) which has now been updated with new results (v3).
>
> * In addition, we’d like to clarify how we calculated the 7.2% and 14.1% performance gains on UCF101 and HMDB51. We obtained these numbers by comparing our best model (Tab 2. 94.1% UCF; 67.2% HMDB) with the fully supervised model from Table 1 (86.9% UCF; 53.1% HMDB) that we trained ourselves using the same backbones. We acknowledge that this may not be a fair comparison because they are pretrained on different datasets, i.e., Kinetics-Sounds (w/ labels) for the supervised model and AudioSet (w/o labels) for the self-supervised model. To avoid confusion, we have removed this claim from the paper as well.

---

### Official Review · AnonReviewer4 · 2020-10-27
**Nice trick, well-motivated and supported by results. Clear improvement over prior work. Text is quite repetitive though making it a boring read unfortunately**

**Rating:** 7
**Confidence:** 5

**Review:**

In addition to my review - directly next, I also give this paper to a junior PhD student who I believe did a very good job in the review. There's no disagreement between both reviews, and I have checked both for correctness, so I provide both of them below: [so it's a 2-in-1 review option.]

================

The paper is straight-forward, easy to read and clearly written. It builds on the Momentum Contrast (MoCo) paper (2020), which aims for a moving average dictionary per modality along with cross-modal contrastive learning for self-supervised tasks, evaluated on audio-visual datasets. The paper takes this a step further by arguing for a more active way to select negative samples that form the dictionary (or negative memory). While dictionaries are efficient ways to perform contrastive learning, this paper showcases that the random selection of negative samples can have a substantive impact on the performance.
To resolve this, the paper proposes to select negatives based on two criteria: informativeness measured by the gradients of the sample and diversity (or representativeness of the potential classes) through a greedy (or online) K-means approach. When both are utilised, the result showcases state-of-the-art on downstream recognition tasks on 3 standard datasets. Results demonstrate the effectiveness of the approach.

The paper is scientifically correct and the proposal is explained both theoretically, with equations, with an algorithm and supported by results.

My only concern with the paper is that by the end of the introduction, the proposal and concept are already well-stated. The rest of the paper keeps repeating the same argument. Apart from the introduction, only 3.3 offers more details. Related work, sections 3.1 and 3.2, figure and algorithm all repeat the same thing in various forms. While emphasis is good, I found the repetitiveness to affect my interest in reading the paper. I recommend that the authors revisit their manuscript to remove the repetitive argument (e.g. the notion that you need both informative and diverse negative samples is made in 4 different locations).
Instead, the authors can provide more insight into the negative dictionary for the variety of models and datasets, showcasing the impact on certain classes. They can also discuss failure cases and whether they relate to classes that are still under-represented in active negative sampling. Comparing an online to offline diversity options (to know the upper limit) would also be interesting.

Additionally, the need for fully connected layers (described in page 6) is only showcased for XMoCo, and not for the final proposal. An ablation of its effect on the final proposal would be important to include.

==============

Summary

This paper proposes an approach to actively sample negative examples for audio-visual representation learning and combines this with an audio-visual approach for contrastive learning. The paper suggests that an active sampling approach that samples datapoints that are are both diverse and uncertain would result in a much more informative dictionary of negative samples, and uses gradient information and the k-MEANS++ algorithm to create such a dictionary of negatives. They then perform a cross-modal dictionary lookup where the queries and keys for each modality are each represented by a learnt encoder where the parameters of the query encoders are learnt directly via backpropogation and the key encoders' parameters are defined as a moving average of the query encoders' parameters. A contrastive loss is then used to learn a representation that is effective for fine-tuning to down-stream tasks, achieving state-of-the-art results for both action recognition and audio event classification.

Strengths

The paper presents novel, well motivated results in the field of cross-modal representation learning. It includes detailed comparison to the state-of-the-art across a wide selelection of papers and datasets. The proposed method shows state-of-the art results across small, medium and large pre-trained datasets for both action recognition and audio even classification.

Weaknesses

Whilst there are detailed comparisons between related works the paper does not discuss many of these related works in the related work section. It would be nice to have some more information on the related works, how they relate, how they differ and the gap that this work fits into.
Additionaly, some information on the models used for audio and visual encoders would help with understanding. They are described in the appendix but there is no reference to them in the main text. The paper adopts a style wherein the entire process is shown as a single algorithm, and the text describes the algorithm step-by-step. This is, in general, quite clear, however continually changing pages to check the algorithm can decrease readability. Perhaps putting the full algorithm in the appendices and providing a simplified version in the main body would be clearer, or alternatively, splitting the algorithm in to multiple, smaller components.
The comparisons to related work are quite detailed, however, in some cases the proposed method offers only a marginal improvement (particularly for the medium and large datasets). Running some experiments multiple times and including error bars would strengthen the state-of-the-art claims. The comparisons could also be improved by consistently comparing across the same architectures. For example, the proposed method is only trained with R(2+1)D-18 for the large dataset, despite the architecture being widely used within the medium dataset comparisons. Perhaps two or three architectures across each dataset size would further strengthen comparisons.

Further points:
*) typo: "meduim scale"
*) Random overleaf link in the appendix
*) references to relevant sections in the appendix would be nice

---

> ### Author Response · Authors · 2020-11-21
> **Appreciate your positive feedback! We improved the writing to make it a little more enjoyable.**
>
> We appreciate the reviewer’s detailed and constructive feedback! We have revised our paper with the following changes:
>
> * We have removed some of the repetitive arguments made from the paper (especially the one on the need for informative and diverse negative samples).
>
> * With regarding to your suggestion of showing more insights and analysis, we have added discussion about negative sampling approaches in Related Work; we have added Table 2 to show the different effects of sampling through Gradient embedding vs. Feature embedding; we have updated Figure 5 to show the relationship of the gradient magnitude with the negative samples.
>
> * We have updated Table 1 that shows the importance of fully-connected layers in our final proposal (and not just on XMoCo).
>
> * We have simplified Algorithm 1 and moved the original version to the Appendix.
>
> * We have moved information on the models used for audio and visual encoders from the Appendix to the main paper.

---

### Official Review · AnonReviewer3 · 2020-10-27
**Nice idea and decent results, lacks discussion of other existing hard negative mining strategies**

**Rating:** 6
**Confidence:** 4

**Review:**

The goal of the paper is audio-visual self-supervised learning using an active sampling technique to mine hard negatives during training.

Strengths
- The paper is well written and clearly explained.
- The strategy for selecting negatives based on diversity seems well reasoned and experimentally outperforms random sampling and OHEM. I like the study showing empirically more categories covered in active sampling vs random sampling.

Weaknesses:
- It would be nice to see some discussion of the other self-supervised contrastive works that also focus on the optimal selection of “negatives”: eg. Korbar et al 2018 (where they use a curriculum based on time distance of negatives from the positives), Iscen et al. 2018: https://arxiv.org/abs/1803.11095, Cao et al.2020: https://arxiv.org/abs/2006.14618, Wu et al 2020: https://arxiv.org/abs/2005.13149 (variational extension to InfoNCE with modified strategies for negative sampling). In a similar vein, the comparison to OHEM in the supplementary is quite nice and I believe should be in the main paper.
- Why is the ablation showing the benefits of Active Sampling only on Kinetics-Sound? Would the same trends shown in Table 1 hold on a random sampled proportion of Kinetics of the same size, or even on the whole dataset?
- The audio-visual contrastive method has been proposed in numerous papers before, so the novelty of this paper lies solely in the active sampling technique that increases diversity using K-means clustering
- (Minor) The footnote on page 1 is a bit confusing, it’s hard to see the probability drop from just Fig. 2 since the division needs to be done, probably better to plot the probabilities directly somewhere or remove this.
- (Minor) It would be interesting to also test the audio representations for classification on VGG-Sound (http://www.robots.ox.ac.uk/~vgg/data/vggsound/)

---

> ### Author Response · Authors · 2020-11-21
> **Thanks! We added discussion of existing negative mining approaches**
>
> We appreciate the reviewer’s positive comments and constructive feedback to improve our paper. We have revised our paper by incorporating your suggestions, including:
>
> * We have added discussion of the four references (Korbar et al., 2018, Iscen et al. 2018, Cao et al., 2020, and Wu et al., 2020) in Section 4.
>
> * We have moved the comparison with OHEM from the Appendix to the main paper; see Table 3.
>
> * We chose Kinetics-Sounds for our ablation study because the dataset is relatively “clean” with a high probability of audio-visual correspondence. Any effect we see in this dataset will transfer to Kinetics and AudioSet, but perhaps to a lesser degree due to the real-world noise. We clarified this in our paper. There is also a practical challenge of conducting ablations at a large-scale due to the heavy GPU requirement.
>
> * We have removed footnote 1 and added a new figure that directly plots the probabilities; see Figure 4 in the Appendix.

---

### Official Review · AnonReviewer1 · 2020-10-29
**Good results, but the manuscript is difficult to get into**

**Rating:** 7
**Confidence:** 3

**Review:**

This paper proposes a cross modal (audio and video) contrastive learning scheme to pretrain on one of the modalities. They also propose a sampling scheme for the negative examples.

The experiments and the corresponding ablations suggest that the proposed methods is helping overall. The table 1 suggests that adding the proposed components (cross-modal pretraining, active sampling for negative examples) seem to help overall.

Also, the samples in the appendix seem to suggest that the proposed `'active negative sampling``'  scheme seems to provide more relevant negative samples. Figure 2 in the experiments also show that the proposed negative sampling mechanism seem to increase the diversity in the negative samples.  Figure 2, also showcases that the proposed negative sampling scheme works better than picking out examples by looking at feature embeddings. This suggests that the proposed negative sampling scheme based on looking at gradients is advantageous over a more standard feature-level approach.

I am not entirely sure if table 2 is exhaustive on the state-of-the-art (SOTA), but from what I gather from the narrative of the paper, they seem to improve the SOTA numbers on three different pretraining dataset size regimes for action recognition datasets. They also provide a similar experiment for audio event classification in table 3, and claim SOTA results. (where they pretrain on different datasets, and test on another)

I think the clarity of the manuscript could be improved, especially in Section 3.3. I think that perhaps you could replace the low level algorithm with a higher-level pseudocode or a diagram and take the algorithm to the supplementary materials?

Also, my understanding of the proposed dictionary building / sampling approach for negative sampling is that you look at the magnitude of the gradients with respect to the contrastive learning objective, and pick the examples which give a large gradient magnitude. I think that this could be made clearer for the reader, perhaps by including a toy example, or including a figure in the appendix which showcases the gradient magnitude vs. the chosen negative examples.

---

> ### Author Response · Authors · 2020-11-21
> **Truly appreciate your positive comments. We have improved the clarify of the manuscript in the revision.**
>
> We appreciate the positive feedback and suggestions to improve our paper! We have incorporated your suggestions into our revised paper, including:
>
> * Algorithm 1: We replaced the low-level equations with high-level pseudo-code and moved the original version with the full details to the Appendix.
>
> * We have added additional results to Figure 5 (in Appendix) showcasing the gradient magnitude vs. the chosen negative examples. The results clearly show that hard negatives tend to have larger gradient norms than easy negatives. It provides the evidence showing that gradient magnitude is an effective measurement of the uncertainty of the current model. We add a more detailed analysis in Appendix C.

---

### Public Comment · ~Yonglong_Tian1 · 2020-11-10
**Good work, but missing related work**

Hi, Authors,

Congrats on this nice work!

One tiny comment: while you claim cross-modal contrastive learning, I wonder if the paradigm in the following paper is well related?

Tian et. al. "Contrastive Multiview Coding", ECCV 2020.

In CMC, we also perform cross-modal/view contrastive learning in a wide range of settings.

---

> ### Author Response · Authors · 2020-11-12
> **Thanks**
>
> Hi Yonglong, thanks for your suggestion.
> Yes, we do think CMC is a good work and it is well related with ours. We will incorporate it in our related work.

---

### Decision · Program_Chairs · 2021-01-07
**Final Decision**

**Decision:**

Accept (Poster)

**Comment:**

The paper focuses on the task of learning audiovisual representations through contrastive learning on unlabelled videos. This work is another addition to the ever-growing literature on self-supervised learning (SSL) with emphasis on video and multi-modal data. The main contribution of this work is the manner in which it tackles a well-known drawback of contrastive learning, namely the strategy used to sample negatives in the contrastive pipeline. The authors propose an active sampling strategy that adaptively chooses negative samples that are informative and diverse. This active selection technique is similar in spirit to many selector functions proposed in the active learning literature. It seems to be the first time it is used for contrastive SSL.

Based on all the reviews and the subsequent discussions, it seems that the reviewers' comments were addressed. The authors are commended on integrating the reviewers' suggestions and making the necessary edits to the paper in a timely manner.